# RIS-Assisted High-Speed Communications with Time-Varying Distance-Dependent Rician Channels

Ke Wang , Chan-Tong Lam * and Benjamin K. Ng

Faculty of Applied Sciences, Macao Polytechnic University, Macao 999078, China
* Correspondence: ctlam@mpu.edu.mo

**Abstract:** Reconfigurable intelligent surface (RIS) has been envisioned as one of the promising solutions for enhancing signal transmissions in high-speed communications (HSC). In this paper, we present a time-varying channel model with distance-dependent Rician factors for the RIS-assisted HSC. Our model not only contains Rayleigh components and Doppler shift (DS) terms but also distance-dependent Rician factors, for characterizing time-varying features. In particular, we show that when the vehicle is far from the base station and the RIS, the channel contains only Rayleigh fading. However, when they are close enough, the channel can be considered as a light-of-sight channel. Based on the proposed model, it is proven that using RIS phase shift optimization, the DS of the cascaded links can be aligned with the DS of the direct link; and if the direct link is blocked, the DS can be removed entirely. Furthermore, we derive the closed-form expressions for the ergodic spectral efficiency and the outage probability of the proposed system. Besides, it is observed that the deployment strategy also affects the system performance. Simulation results validate all analyses.

**Keywords:** reconfigurable intelligent surfaces; Doppler effects; high-speed communications; distance-dependent Rician factors

## 1. Introduction

Reconfigurable intelligent surface (RIS), also known as large intelligent surface, and intelligent reflecting surface [1], is a tunable surface that consists of massive passive elements at sub-wavelengths that can change the amplitude and phase of the incident signal independently in real-time [2]. Due to its capability of creating and controlling the smart radio environment, RIS has already demonstrated its potential in many areas of wireless communication, such as localization [3] and coverage enhancement [4]. In particular, Pan et al. [2] provided the comprehensive overview of recent advances in RIS-assisted wireless communications from the signal processing perspective, which showed the ability to control electromagnetic waves by the RIS. Keykhosravi et al. [3] consider a problem of positioning a moving user in a three-dimensional space based on the received signal from a single-antenna transmitter and reflected signal from an RIS. Ibrahim et al. [4] developed a framework based on moment generation functions to characterize the coverage probability of a user in RIS-assisted wireless systems, which revealed the potential of the RIS on transmission coverage area enhancement.

High-speed communications (HSC) characterize high velocities of the transceivers, along with significant permeation losses of signals [5]. Therefore, there has been a growing interest in HSC with RIS recently to deal with some problems such as Doppler mitigation, received power maximization, and coverage area enhancement [6–10]. In particular, Huang et al. [7] proposed a new high-mobility communication system that utilizes the signal reflection capability of the RIS and employed it with vehicle-to-everything communications. For high-mobility scenarios with doubly selective Rician fading, Xu et al. [8] presented a fully developed RIS-aided transmission system. Considering RIS-aided millimeter wave HSC systems, transmission beamforming at the base station (BS) and phase

shifts at the receiver have been investigated in [9] for the purpose of maximizing spectral efficiency (SE). Besides, Wang et al. [10] revealed that with the RIS, the SE and the Doppler effect can be respectively maximized and minimized simultaneously. The authors of [11] investigated the impact of hardware impairments (HWI) on the performance of Doppler mitigation by the RIS in HSC. Interestingly, they showed that the HWI enlarges the delay spread of the HSC, but not the ability to mitigate Doppler effects. Moreover, the authors of [12] showed that due to the HWI, a multiple RIS system suffers from a wider delay spread as well as significant degradation of the SE. However, the channel models considered in [10–12] did not take into account the non-light-of-sight (NLoS), i.e., the Rayleigh component.

Besides, it is noteworthy that [7–12] did not consider distance-dependent Rician factors, which is not practical in HSC systems. This is because the distances of the direct and the cascaded links vary dramatically considering the mobility of the transceiver [6,13]; it is more practical to include the distance-dependent Rician factor in the channel model for HSC systems. In particular, when the transceivers are far away from one another, the NLoS part dominates the channel, but the light-of-sight (LoS) channel leads when they are close. Matthiesen et al. [14] proposed a continuous time-varying channel model for a satellite communication system with the RIS. This model, however, did not consider the NLoS part. Based on the model in [14], Björnson et al. [15] revealed that the Doppler shift (DS) cannot be removed if the direct link exists in an RIS-assisted communication system. However, if the direct link is blocked, the DS can be mitigated totally. Basar [16] showed that the Doppler effect would disappear if there were enough RISs deployed around the scattering environment, but the proposed model is too simple since the author assumed every RIS in the system only contains one reflective element, which cannot beamform the incident signal in reality. Therefore, we consider more practical channel modeling for the RIS-assisted HSC system in this paper.

A geometry-based stochastic model (GBSM) has been applied for modeling the RIS-assisted communication system recently [17,18]. Its computational complexity, however, precludes subsequent research. Similarly, some models are so simple that they cannot reflect the characteristics of the HSC system with RIS [10,14–16]. Given this background, in this paper, we first propose a time-varying model for the RIS-aided HSC system containing the NLoS part, the DS term, and the distance-dependent Rician factors. We replace the empirical constants with distance-dependent linear expressions to define the dynamic Rician factors and then analyze the SE and outage probability of the proposed system. Table 1 compares and contrasts our contributions to state-of-the-art RIS-assisted HSC studies. In particular, Refs. [10,11,14,15] did not consider the NLoS part, the DS term, or the distance-dependent Rician factor; Refs. [12,16,19] ignored the DS term and the distance-dependent Rician factor. Besides, although Refs. [7,8] considered the NLoS part and the DS term, they used fixed Rician factors, which are impractical. Moreover, in [17,18], the authors used the GBSM method to model the channel. However, they also ignored the dynamic Rician factors, and the modeling complexities are very high.

**Table 1.** State-of-the-art time-varying RIS-assisted HSC models.

| | Consider the NLoS Part? | Consider the DS Term? | Consider the Distance-Dependent Rician Factor? | Modeling Complexity |
|---|---|---|---|---|
| [10,11,14,15] | No | No | No | Low |
| [12,16,19] | Yes | No | No | Low |
| [7] | Yes | Yes | No | Moderate |
| [8] | Yes | Yes | No | High |
| [17,18] | Yes | Yes | No | High |
| Ours | Yes | Yes | Yes | Low |

The major contributions of this paper are as follows:

- We propose a time-varying channel model for RIS-assisted HSC systems. Compared to previous works, our model contains NLoS (or Rayleigh fading) components, distance-dependent Rician factors, and DS terms;
- We show that when the moving user is far from the BS and the RIS, the Rayleigh fading dominates the channel. However, when they are close, the channel is almost equivalent to the LoS channel;
- For the proposed channel model, we further show that using RIS phase shift optimization, the DS of the cascaded links can be aligned to the DS of the direct link. Furthermore, the DS can be completely eliminated when the direct link is obstructed;
- We derive the closed-form expressions for the ergodic spectral efficiency and outage probability of the proposed system. Besides, it is shown that the deployment strategy has an impact on the system performance.

The rest of this paper is arranged as follows. We present the time-varying channel model with dynamic Rician factors for RIS-assisted HSC in Section 2. Based on the proposed system, we optimize the phase shift of the RIS to maximize the received power and show how the RIS aligns with the DS of the cascaded links to the DS of the direct link in Section 3. In Section 4, we derive the analytical expression for the ergodic SE and outage probability. In Section 5, we provide numerical simulations and discussions, followed by the conclusions and future works in Section 6.

Notation: $\lceil \cdot \rceil$ and $\lfloor \cdot \rfloor$ denote the ceiling and the floor of the argument, respectively. $\mathbb{Z}^+$ is the set of positive integers, $|\cdot|$ refers to the absolute value, $\|\cdot\|$ represents the $l_2$ norm, Prob denotes the probability, $[\cdot]^T$ is the transpose operation for a vector, $(\cdot)^*$ is the conjugate operation for a complex number, $\mathbb{E}\{\cdot\}$ and $\mathbb{V}\{\cdot\}$ denote respectively the expectation and variance operation. Moreover, $\mathcal{CN}(\mu, \sigma^2)$ is the complex Gaussian distribution with expectation $\mu$ and variance $\sigma^2$, $\mathfrak{R}\{\cdot\}$ is the real part of a complex number, and ($a \mod b$) is the remainder of the division of $a$ by $b$. Lastly, $\max\{\cdot\}$ and $\min\{\cdot\}$ denote the maximum and minimum values, respectively.

## 2. System Model with Time-Varying Distance-Dependent Rician Channels

We assume a downlink single-input single-output RIS-assisted HSC system, as shown in Figure 1. There are one roadside BS, one train-side moving mobile relay (MR) with speed $v$, and one RIS with $M$ elements near the BS. Both the BS and the RIS are surrounded by certain obstacles, such as buildings and trees [20,21]. The single antennas of the transceiver and the $M$ elements of the RIS are all isotropic. The origin point of the Cartesian coordinate system is $D_0 = [0, 0, 0]^T$. The positions of the BS, the MR, and the *mn*-th element of the RIS are denoted as $D_{\mathrm{BS}} = [x_{\mathrm{BS}}, y_{\mathrm{BS}}, z_{\mathrm{BS}}]^T$, $D_{\mathrm{MR}}(t) = [x_0 + v \cdot t, y_{\mathrm{MR}}, z_{\mathrm{MR}}]^T$ and $D_{mn}$, respectively. Besides, we assume the channel state information can be obtained in advance using the predictive information [10,11].

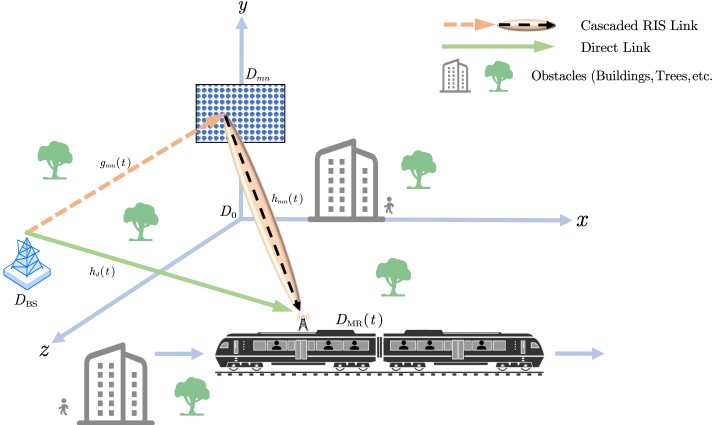

**Figure 1.** An RIS-assisted HSC system. Note that the obstacles cause NLoS components for direct and cascaded links.

In the proposed HSC scenario, large signal penetration losses can be avoided by establishing the MR on the top of the vehicle [5,6]. Besides, the channel model contains two main links. The first one is the direct link $h_d(t)$, which represents the channel between the BS and the MR. The second one is the cascaded RIS link, which means the channels from the BS and the RIS $g_{mn}(t)$, and from the RIS to the MR $g_{mn}(t)$. Note that both $g_{mn}(t)$ and $g_{mn}(t)$ include LoS and NLoS components. Moreover, in this paper, we assume that the proposed channel experiences block fading [7,14,15,19].

*2.1. Direct Link $h_d(t)$ from the BS to the MR*

First, let us consider the LoS part of $h_d(t)$. The channel gain of the direct link is given as [22]

$$A_0(t) = \frac{\lambda}{4\pi d_0(t)}, \tag{1}$$

where $d_0(t) = \|D_{\text{MR}}(t) - D_{\text{BS}}\|$, and $\lambda$ is the carrier wavelength. The delay of the direct link can be calculated as

$$\tau_0(t) = \frac{d_0(t)}{c}, \tag{2}$$

where $c$ is the speed of light. Taking the derivative of $\tau_0(t)$, we have the Doppler frequency shift for $h_d(t)$ as [14]

$$f_0(t) = -f_c \frac{d(\tau_0(t))}{dt} = -f_c \frac{v(x_0 + v \cdot t - x_{\text{BS}})}{c\|D_{\text{MR}}(t) - D_{\text{BS}}\|} = -f_c \frac{v(x_0 + v \cdot t - x_{\text{BS}})}{cd_0(t)} \tag{3}$$

where $f_c$ is carrier frequency. Consider the transmitted passband signal for the MR during time slot $t \in \{1, \ldots, T\}$ as

$$s(t) = \sqrt{2}\Re\{x(t)\exp(\jmath 2\pi f_c t)\}, \tag{4}$$

and $x(t) = x_r(t) + \jmath x_i(t)$, $\mathbb{E}\{|s(t)|^2\} = 1$. $x_r(t)$ and $x_i(t)$ are baseband signals bandlimited to $B/2$, with $B$ denoting bandwidth such that $B \ll f_c$. Thus, the received signal in the MR is $s(t - \tau_0(t))$. The phase of the LoS part of the $h_d(t)$ can be calculated as

$$h_d^{\text{LoS}}(t) = \exp\left(-\jmath 2\pi(f_c + f_0(t))\tau_0\right), \tag{5}$$

where $\jmath$ denotes an imaginary unit.

The NLoS part of $h_d(t)$, which is denoted by $h_d^{\text{NLoS}}(t)$, can be modeled as a random variable following a complex Gaussian distribution $\mathcal{CN}(0,1)$ in a time slot $t$. Let $\kappa_d^{\text{LoS}}$ and $\kappa_d^{\text{NLoS}}$ be respectively the Rician factors for $h_d^{\text{LoS}}(t)$ and $h_d^{\text{NLoS}}(t)$. The direct channel $h_d(t)$ can be obtained as [19]

$$h_d(t) = A_0(t)\left(\sqrt{\frac{\kappa_d^{\text{LoS}}}{\kappa_d^{\text{LoS}} + 1}}h_d^{\text{LoS}}(t) + \sqrt{\frac{1}{\kappa_d^{\text{NLoS}} + 1}}h_d^{\text{NLoS}}(t)\right). \tag{6}$$

**Remark 1.** *It is important to point out that (6) is different from the direct link in the previous works [10,11,14,15], which did not consider the Rician factors $\kappa_d^{\text{LoS}}$ and $\kappa_d^{\text{NLoS}}$, NLoS channel part $h_d^{\text{NLoS}}(t)$, and DS term $f_0(t)$.*

*2.2. Cascaded RIS Link $h_{\text{RIS}}(t)$ from the BS to the RIS and from RIS to the MR*

Suppose the RIS is a planar surface on a rectangular grid spaced $dx$ and $dy$ apart in the three-dimensional Cartesian coordinate system. Assume the size and the geometric center of

the RIS in Figure 1. are $\sqrt{M} \times \sqrt{M}$ and $[0, r, 0]^{\mathrm{T}}$, respectively. The center of the *mn*-th element is given as

$$D_{mn} = \left[ \Psi(m, dx, \sqrt{M}), \Psi(n, dy, \sqrt{M}) + r, 0 \right]^{\mathrm{T}}, \tag{7}$$

where $\Psi(m, dx, \sqrt{M}) = dx(m - 0.5((\sqrt{M} + 1) \bmod 2)), m \in \Omega(\sqrt{M})$ with

$$\Omega(\sqrt{M}) = \left\{ ((\sqrt{M} + 1) \bmod 2)) - \left\lfloor \frac{\sqrt{M}}{2} \right\rfloor, \dots, \left\lfloor \frac{\sqrt{M}}{2} \right\rfloor \right\}, \tag{8}$$

and $\Psi(n, dy, \sqrt{M})$ is obtained in the same way as $\Psi(m, dx, \sqrt{M})$.

Assume that the BS-RIS and the RIS-MR links, i.e., $g_{mn}(t)$ and $h_{mn}(t)$, both have LoS and NLoS parts. Let us consider the LoS part of $h_{\mathrm{RIS}}(t)$ first. For the *mn*-th isotropic element of the RIS, the channel gain of the BS-RIS and the RIS-MR links are given respectively as

$$A_{\mathrm{BS}}^{mn} = \frac{\lambda}{4\pi d_{\mathrm{BS}}^{mn}} \tag{9}$$

and

$$A_{mn}^{\mathrm{MR}}(t) = \frac{\lambda}{4\pi d_{mn}^{\mathrm{MR}}(t)}, \tag{10}$$

where $d_{\mathrm{BS}}^{mn} = \|D_{mn} - D_{\mathrm{BS}}\|$ and $d_{mn}^{\mathrm{MR}}(t) = \|D_{\mathrm{MR}}(t) - D_{mn}\|$. The delay of the *mn*-th cascaded link is

$$\tau_{mn}(t) = \tau_{\mathrm{BS}}^{mn} + \tau_{mn}^{\mathrm{MR}}(t) = \frac{d_{\mathrm{BS}}^{mn}}{c} + \frac{d_{mn}^{\mathrm{MR}}(t)}{c}. \tag{11}$$

The DS for the *mn*-th cascaded link $h_{mn}(t)$ can be calculated as

$$\begin{aligned}
f_{mn}(t) &= -f_c \frac{d}{dt} \left( \tau_{mn}^{\mathrm{MR}}(t) + \frac{\phi_{mn}(t)}{2\pi f_c} \right) \\
&= -f_c \left( \frac{v(x_0 + vt - x_{mn})}{c d_{mn}^{\mathrm{MR}}(t)} + \frac{d}{dt} \left( \frac{\phi_{mn}(t)}{2\pi f_c} \right) \right),
\end{aligned} \tag{12}$$

where $\phi_{mn}(t) \in (0, 2\pi]$ is the tunable phase shift of the *mn*-th element. We assume the maximal amplitude of 1 for all of the elements.

Therefore, the *mn*-th RIS-MR link, $h_{mn}(t)$, can be obtained as

$$h_{mn}(t) = A_{mn}^{\mathrm{MR}}(t) \left( \sqrt{\frac{\kappa_{mn}^{\mathrm{LoS}}}{\kappa_{mn}^{\mathrm{LoS}} + 1}} h_{mn}^{\mathrm{LoS}}(t) + \sqrt{\frac{1}{\kappa_{mn}^{\mathrm{NLoS}} + 1}} h_{mn}^{\mathrm{NLoS}}(t) \right), \tag{13}$$

where

$$h_{mn}^{\mathrm{LoS}}(t) = \exp\left( -\jmath 2\pi (f_c + f_{mn}(t)) \tau_{mn}^{\mathrm{MR}}(t) \right), \tag{14}$$

and $h_{mn}^{\mathrm{NLoS}}(t)$ is the same as $h_d^{\mathrm{NLoS}}(t)$, $\kappa_{mn}^{\mathrm{LoS}}$ and $\kappa_{mn}^{\mathrm{NLoS}}$ respectively denote Rician factors of the LoS and NLoS parts of $h_{mn}(t)$. Similarly, the *mn*-th BS-RIS link, $g_{mn}(t)$, can be obtained as

$$g_{mn}(t) = A_{\mathrm{BS}}^{mn} \left( \sqrt{\frac{\rho_{mn}^{\mathrm{LoS}}}{\rho_{mn}^{\mathrm{LoS}} + 1}} g_{mn}^{\mathrm{LoS}} + \sqrt{\frac{1}{\rho_{mn}^{\mathrm{NLoS}} + 1}} g_{mn}^{\mathrm{NLoS}}(t) \right), \tag{15}$$

where $g_{mn}^{\mathrm{LoS}} = \exp(-\jmath 2\pi f_c \tau_{\mathrm{BS}}^{mn})$, $\rho_{mn}^{\mathrm{LoS}}$ and $\rho_{mn}^{\mathrm{NLoS}}$ are Rician factors of the LoS and NLoS channels of $g_{mn}(t)$ respectively, and $g_{mn}^{\mathrm{NLoS}}(t) \sim \mathcal{CN}(0, 1)$. Note that although the $g_{mn}^{\mathrm{LoS}}$ in (15) is not a function of time $t$, $g_{mn}(t)$ is still time-dependent because the NLoS part,

i.e., $g_{mn}^{\text{NLoS}}(t)$, would change due to moving obstacles such as cars and pedestrians. Finally, we have the cascaded link as

$$h_{\text{RIS}}(t) = \sum_{m=1,n=1}^{\sqrt{M},\sqrt{M}} h_{mn}(t) \exp\left(-\jmath\phi_{mn}(t)\right) g_{mn}(t). \tag{16}$$

### 2.3. Distance-Dependent Rician Factors

The Rician factors $\kappa_d^{\text{LoS}}$, $\kappa_d^{\text{NLoS}}$, $\rho_{mn}^{\text{LoS}}$, $\rho_{mn}^{\text{NLoS}}$, $\kappa_{mn}^{\text{LoS}}$ and $\kappa_{mn}^{\text{NLoS}}$ in (6), (13) and (15) are all time-invariant, which is not practical for HSC systems, especially when the BS and the RIS are surrounded by obstacles. Therefore, we use the distance-dependent Rician factor $\kappa(t)$ [20] in the proposed channel model, given as

$$\kappa(t) = 10^{0.1(\varrho - \iota \cdot d(t))}, \tag{17}$$

where $\varrho > 0$ and $0 < \iota < 1$ are environmental parameters and $d(t)$ denotes the distance between the transmitter and receiver in meters. Thus, we have $\kappa_d^{\text{LoS}}(t) = \varrho - \iota \cdot d_0(t)$, and other Rician factors can be obtained in the same way. Consequently, (6), (13) and (15) can be rewritten as

$$h_d(t) = A_0(t)\left(\sqrt{\frac{\kappa_d(t)}{\kappa_d(t)+1}} h_d^{\text{LoS}}(t) + \sqrt{\frac{1}{\kappa_d(t)+1}} h_d^{\text{NLoS}}(t)\right), \tag{18}$$

$$h_{mn}(t) = A_{mn}^{\text{MR}}(t)\left(\sqrt{\frac{\kappa_{\text{RIS}}(t)}{\kappa_{\text{RIS}}(t)+1}} h_{mn}^{\text{LoS}}(t) + \sqrt{\frac{1}{\kappa_{\text{RIS}}(t)+1}} h_{mn}^{\text{NLoS}}(t)\right), \tag{19}$$

and

$$g_{mn}(t) = A_{\text{BS}}^{mn}\left(\sqrt{\frac{\rho_{\text{RIS}}}{\rho_{\text{RIS}}+1}} g_{mn}^{\text{LoS}} + \sqrt{\frac{1}{\rho_{\text{RIS}}+1}} g_{mn}^{\text{NLoS}}(t)\right), \tag{20}$$

where $\kappa_d(t) = \kappa_d^{\text{LoS}}(t)$, $\kappa_{\text{RIS}}(t) = \frac{1}{M}\sum_{m=1,n=1}^{\sqrt{M},\sqrt{M}} \kappa_{mn}^{\text{LoS}}(t)$, and $\rho_{\text{RIS}} = \frac{1}{M}\sum_{m=1,n=1}^{\sqrt{M},\sqrt{M}} \rho_{mn}^{\text{LoS}}$. Based on (18)–(20), the total channel $h(t)$ can be written as

$$h(t) = h_d(t) + h_{\text{RIS}}(t) \tag{21}$$

$$= h_d(t) + \sum_{m=1,n=1}^{\sqrt{M},\sqrt{M}} h_{mn}(t) \exp\left(-\jmath\phi_{mn}(t)\right) g_{mn}(t).$$

Note that the time delay of the $mn$-th element of the RIS itself is $\phi_{mn}(t)/(2\pi f_c)$. Therefore, the received signal in the $mn$-th cascaded link, $s(t)$, is given as $s\left(t - \tau_{mn}(t) - \frac{\phi_{mn}(t)}{2\pi f_c}\right)$. Consequently, the received signal $y(t)$ can be written as

$$y(t) = \sqrt{P_t}\Bigg( h_d(t) s\left(t - \tau_0(t)\right)$$

$$+ \sum_{m=1,n=1}^{\sqrt{M},\sqrt{M}} h_{mn}(t) \exp\left(-\jmath\phi_{mn}(t)\right) g_{mn}(t) s\left(t - \tau_{mn}(t) - \frac{\phi_{mn}(t)}{2\pi f_c}\right)\Bigg) + w(t), \tag{22}$$

where $P_t$ is transmitted power and $w(t)$ is an additive white Gaussian noise with power $\sigma^2$ in a time slot $t$.

### 3. Received Power Maximization and Doppler Shift Alignment

In this section, we first obtain the optimal phase shift set of the RIS, based on the proposed channel model. Then, we demonstrate how the RIS can be used to align the DS. The results reveal that when the transceiver and RIS are close, the LoS channel dominates and vice versa; and optimizing the phase shift set cannot remove the DS unless the direct link vanishes.

### 3.1. Received Power Maximization

Using (22), the average received signal power can be calculated as $\mathbb{E}\{|y(t) - w(t)|^2\}$. In order to maximize it, the phase parts of the LoS components for the direct and cascaded links should be the same. Thus, we have $\phi_{mn}(t) = 2\pi f_c(\tau_0(t) - \tau_{mn}(t))$. However, since $\tau_0(t) \geq \tau_{mn}(t)$, then $\phi_{mn}(t) \leq 0$, which cannot be achieved based on our model. Thus, $\phi_{mn}(t)$ should be added by an additional full carrier signal period delay $2\pi\zeta_{mn}(t)$ to ensure $\phi_{mn}(t) \in (0, 2\pi]$. Therefore, the continuous optimal phase shift set for maximizing the received signal power is

$$\phi_{mn}^{\mathrm{opt}}(t) = 2\pi f_c\left(\tau_0(t) - \tau_{mn}(t)\right) + 2\pi\zeta_{mn}(t), \tag{23}$$

where $\zeta_{mn}(t) = \left\lceil f_c(\tau_{mn}(t) - \tau_0(t)) \right\rceil$. It is noteworthy that (23) can not only maximize $\mathbb{E}\{|y(t) - w(t)|^2\}$ but also minimize the delay spread of the whole system [11,14].

**Remark 2.** *It is observed that the optimal phase shift set $\phi_{mn}^{\mathrm{opt}}(t)$ in (23) consists only of $f_c$, $\tau_0(t)$ and $\tau_{mn}(t)$. In other words, the RIS works on the LoS part of the channel, instead of the NLoS one. Thus, the RIS would reflect more signals when the MR, the BS, and the RIS are close, i.e., the Rician factor becomes large and the LoS part dominates. Similarly, when $d_0(t)$ and $d_{mn}^{\mathrm{MR}}(t)$ are very large, the system performance would be diminished not only by the decrease of the channel gain $A_0(t)$ and $A_{mn}^{\mathrm{MR}}(t)$ in (6) and (13) but also the distance-dependent Rician factor, which causes the whole channel to be Rayleigh fading only.*

### 3.2. Doppler Shift Alignment

Considering the optimal phase shift $\phi_{mn}^{\mathrm{opt}}(t)$ in (23), recall the DS $f_0(t)$ in (3) and $f_{mn}(t)$ in (12), and note that $d(\zeta_{mn}(t))/dt = 0$, we have

$$\begin{aligned}
f_{mn}(t) &= -f_c\frac{d}{dt}\left(\tau_{mn}^{\mathrm{MR}}(t) + \frac{\phi_{mn}^{\mathrm{opt}}(t)}{2\pi f_c}\right) \\
&= -f_c\left(\frac{v(x_0 - x_{mn} + v\cdot t)}{cd_{mn}^{\mathrm{MR}}(t)} + \frac{d}{dt}\left(\frac{\phi_{mn}^{\mathrm{opt}}(t)}{2\pi f_c}\right)\right) \\
&= -f_c\left(\frac{v(x_0 - x_{mn} + v\cdot t)}{cd_{mn}^{\mathrm{MR}}(t)} + \frac{v(x_0 - x_{\mathrm{BS}} + v\cdot t)}{cd_0(t)} - \frac{v(x_0 - x_{mn} + v\cdot t)}{cd_{mn}^{\mathrm{MR}}(t)}\right) \\
&= -f_c\frac{d(\tau_0(t))}{dt} = f_0(t).
\end{aligned} \tag{24}$$

**Remark 3.** *It can be observed that if the optimal phase shift $\phi_{mn}^{\mathrm{opt}}(t)$ in (23) is adopted, then $f_0(t)$ is equal to $f_{mn}(t)$. In other words, although optimizing the phase shift of the RIS cannot remove the DS entirely, it can align them to $f_0(t)$. Accordingly, the carrier frequency of the received signal $y(t)$ in (22) changes from $f_c$ to $f_c + f_0(t)$. Note that $f_0(t)$ can be ignored if $f_0(t) \ll f_c$.*

Consider the case where the direct link is blocked; we have $h_d(t) = 0$, and $h(t)$ in (21) can be rewritten as

$$\begin{aligned}
h(t) &= h_{\mathrm{RIS}}(t) \\
&= \sum_{m=1,n=1}^{\sqrt{M},\sqrt{M}} h_{mn}(t)\exp\left(-\jmath\phi_{mn}(t)\right)g_{mn}(t).
\end{aligned} \tag{25}$$

Accordingly, the received signal $y(t)$ in (22) can be rewritten as

$$y(t) = \sum_{m=1,n=1}^{\sqrt{M},\sqrt{M}} h_{mn}(t)\exp\left(-\jmath\phi_{mn}(t)\right)g_{mn}(t)s\left(t - \tau_{mn}(t) - \frac{\phi_{mn}(t)}{2\pi f_c}\right) + w(t). \tag{26}$$

Since $\tau_0(t) = 0$, the optimal phase shift $\phi_{mn}^{\text{opt}}(t)$ in (23) is changed to

$$\phi_{mn}^{\text{opt}}(t) = 2\pi f_c\left(-\tau_{mn}(t)\right) + 2\pi \zeta_{mn}(t), \tag{27}$$

where $\zeta_{mn}(t) = \lceil f_c \tau_{mn}(t) \rceil$.

Based on (27), we can compute the DS $f_{mn}(t)$ as

$$
\begin{aligned}
f_{mn}(t) &= -f_c \frac{d}{dt}\left(\tau_{mn}^{\text{MR}}(t) + \frac{\phi_{mn}(t)}{2\pi f_c}\right) \\
&= -f_c\left(\frac{v(x_0 - x_{mn} + v \cdot t)}{c d_{mn}^{\text{MR}}(t)} - \frac{v(x_0 - x_{mn} + v \cdot t)}{c d_{mn}^{\text{MR}}(t)}\right) \\
&= 0.
\end{aligned}
\tag{28}
$$

**Remark 4.** *As (28) shows, when there is no direct link $h_d(t)$, the RIS can not only align the DS but remove the Doppler spread, i.e., $|f_{mn}(t) - f_0(t)| = 0$. This is the so-called Doppler cloaking [15], and the carrier frequency remains to $f_c$ in this case. Note that with the RIS phase shift optimization, the Doppler spread can be removed. However, the DS cannot be totally eliminated unless the direct link vanishes. Thus, DS and Doppler spread are jointly considered in this paper.*

## 4. Performance Analysis

In this section, we first give the analytical expression of the upper bound for the ergodic SE. Then, we analyze the effects of Rician factors when the vehicle is far from (near) the BS and the RIS. Besides, we show that the channel $h(t)$ follows complex-valued Gaussian distribution, and so obtain the outage probability of the $h(t)$. The results reveal that the distance-dependent Rician factor not only affects the ergodic SE but influences the outage probability.

*4.1. Ergodic Spectral Efficiency*

**Proposition 1.** *Consider $P_t$ and $\sigma^2$ are respectively the transmitted and the additional noise power, and $h(t)$ is the channel for the proposed model. Then, the ergodic SE of the RIS-assisted HSC system at time t is upper bounded by*

$$\text{SE}(t) = \mathbb{E}\left\{\log_2\left(1 + \frac{P_t}{\sigma^2}|h(t)|^2\right)\right\} \leq \text{SE}^{\text{upper}}(t) = \log_2\left(1 + \frac{P_t}{\sigma^2}\mathbb{E}\left\{|h(t)|^2\right\}\right), \tag{29}$$

*where*

$$
\begin{aligned}
\mathbb{E}\left\{|h(t)|^2\right\} = {}& \left(K_1^2(t) + K_2^2(t)\right)A_0^2(t) + K_3^2(t)G_1^2\left(\sum_{m=1,n=1}^{\sqrt{M},\sqrt{M}} A_{mn}(t)\right)^2 + \\
& \left(K_3^2(t)K_4^2(t) + K_4^2(t)G_1^2 + K_4^2(t)G_2^2\right)\sum_{m=1,n=1}^{\sqrt{M},\sqrt{M}} A_{mn}^2(t) + \\
& 2K_1(t)K_3(t)G_1 A_0(t)\sum_{m=1,n=1}^{\sqrt{M},\sqrt{M}} A_{mn}(t)
\end{aligned}
\tag{30}
$$

*with $K_1(t) = \sqrt{\frac{\kappa_d(t)}{\kappa_d(t)+1}}$, $K_2(t) = \sqrt{\frac{1}{\kappa_d(t)+1}}$, $K_3(t) = \sqrt{\frac{\kappa_{\text{RIS}}(t)}{\kappa_{\text{RIS}}(t)+1}}$, $K_4(t) = \sqrt{\frac{1}{\kappa_{\text{RIS}}(t)+1}}$, $G_1 = \sqrt{\frac{\rho_{\text{RIS}}}{\rho_{\text{RIS}}+1}}$, $G_2 = \sqrt{\frac{1}{\rho_{\text{RIS}}+1}}$, and $A_{mn}(t) = A_{\text{BS}}^{mn} A_{mn}^{\text{BS}}(t)$.*

**Proof.** See Appendix A. □

**Remark 5.** *If we assume that the gain for all elements is the same, i.e., $A_{mn}(t) = A_1(t)$. This happens when the distance between the BS, the RIS, and the MR is far greater than the size of the RIS [1,2]. Consequently, (30) becomes*

$$\mathbb{E}\{|h(t)|^2\} \approx \left(K_1^2(t) + K_2^2(t)\right) A_0^2(t) + 2MK_1(t)K_3(t)G_1 A_0(t)A_1(t) + $$
$$\left(M^2 K_3^2(t)G_1^2 + M\left(K_3^2(t)K_4^2(t) + K_4^2(t)G_1^2 + K_4^2(t)G_2^2\right)\right) A_1^2(t). \tag{31}$$

**Remark 6.** *If $K_1(t) = K_3(t) = G_1 = 0$, then (30) becomes*

$$\mathbb{E}\{|h(t)|^2\} = K_2^2(t)A_0^2(t) + K_4^2(t)G_2^2 \sum_{m=1,n=1}^{\sqrt{M},\sqrt{M}} A_{mn}^2(t). \tag{32}$$

*And accordingly, (31) becomes*

$$\mathbb{E}\{|h(t)|^2\} \approx K_2^2(t)A_0^2(t) + MK_4^2(t)G_2^2 A_1^2(t). \tag{33}$$

We observe that in this case, $h(t)$ is under Rayleigh fading, and $\mathbb{E}\{|h(t)|^2\}$ is proportional to the number of elements of the RIS, $M$. This often happens when the vehicle is far from the BS and the RIS. Besides, (33) is not related to the phase shift $\phi_m^{\text{opt}}(t)$ of the RIS.

**Remark 7.** *If $K_2(t) = K_4(t) = G_2 = 0$, then (30) becomes*

$$\mathbb{E}\{|h(t)|^2\} = K_1^2(t)A_0^2(t) + K_3^2(t)G_1^2 \left(\sum_{m=1,n=1}^{\sqrt{M},\sqrt{M}} A_{mn}(t)\right)^2 + $$
$$2K_1(t)K_3(t)G_1 A_0(t) \sum_{m=1,n=1}^{\sqrt{M},\sqrt{M}} A_{mn}(t), \tag{34}$$

*and (31) becomes*

$$\mathbb{E}\{|h(t)|^2\} \approx K_1^2(t)A_0^2(t) + M^2 K_3^2(t)G_1^2 A_1^2(t) + 2MK_1(t)K_3(t)G_1 A_0(t)A_1(t). \tag{35}$$

It can be observed that in this case, $h(t)$ only contains LoS components, and $\mathbb{E}\{|h(t)|^2\}$ increases proportional to $M$ and $M^2$. This often appears when the vehicle approaches the BS and the RIS. Besides, it is noteworthy that if the direct link is blocked, we have $\mathbb{E}\{|h(t)|^2\} = A_1^2(t)M^2 K_3^2(t)G_1^2$, which increases proportional to $M^2$, i.e., the *square power law* [15]. Therefore, the gain of the reflected beam from the RIS to the vehicle is more important than that of the direct link because, in practical scenarios, the cascaded link is usually the only channel available, and the square power law can bring a more promising channel gain.

### 4.2. Outage Probability

**Proposition 2.** *The time-varying channel $h(t)$ between BS and MR follows complex-valued Gaussian distribution with mean $\mu_h(t)$ and variance $\sigma_h^2(t)$, namely,*

$$h(t) \sim \mathcal{CN}\left(\mu_h(t), \sigma_h^2(t)\right), \tag{36}$$

*where*

$$\mu_h(t) \triangleq \mathbb{E}\{h(t)\}$$
$$= A_0(t)K_1(t)h_d^{\text{LoS}}(t) + \sum_{m=1,n=1}^{\sqrt{M},\sqrt{M}} A_{mn}(t)\exp\left(-\jmath\phi_{mn}(t)\right)K_3(t)G_1 h_{mn}^{\text{LoS}}(t)g_{mn}^{\text{LoS}}, \tag{37}$$

*and*

$$\sigma_h^2(t) \triangleq \mathbb{V}\{h(t)\} = A_0^2(t)K_2^2(t) + K_4^2(t)G_2^2 \sum_{m=1,n=1}^{\sqrt{M},\sqrt{M}} A_{mn}^2(t). \tag{38}$$

**Proof.** See Appendix B. □

**Remark 8.** *It is shown that the expectation of $h(t)$, i.e., $\mu_h(t)$, relies on the LoS Rician factors, $K_1(t)$, $K_3(t)$, and $G_1$. Therefore, when the MR, the BS, and the RIS are close enough, $\mu_h(t)$ would be maximized. However, the variance $\sigma_h^2(t)$ depends on the NLoS Rician factors, namely, $K_2(t)$, $K_4(t)$, and $G_2$.*

**Proposition 3.** *The outage probability $\mathrm{Prob}_{\mathrm{out}}(t)$ of the $h(t)$ is given by*

$$\mathrm{Prob}_{\mathrm{out}}(t) = 1 - Q_{\frac{1}{2}}\left(\sqrt{\xi(t)}, \sqrt{\gamma_0(t)}\right), \tag{39}$$

*where $\xi(t) = |\mu_h(t)|^2/\sigma_h^2(t)$ and $\gamma_0(t) = \gamma_{\mathrm{th}}/(\bar{\gamma}\sigma_h^2(t))$ with a given SNR threshold $\gamma_{\mathrm{th}}$ and $\bar{\gamma} = P_t/\sigma^2$. Besides, $Q_u(a,b)$ is the Marcum Q-function with the order u.*

**Proof.** See Appendix C. □

**Remark 9.** *It can be seen that $\mathrm{Prob}_{\mathrm{out}}(t)$ is also related to $\mu_h(t)$ in (37) and $\sigma_h^2(t)$ in (38), which means that the distance-dependent Rician factors affect the outage probability of the proposed system. Besides, due to the square power law, increasing the number of elements can bring a lower outage probability compared to increasing the number of antennas in traditional relays.*

## 5. Numerical Evaluations and Discussion

In this section, simulation results are provided to validate the analytical results in Sections 2–4. The top view of the scenario is shown in Figure 2, and unless otherwise specified, all parameters are set to the values in Table 2 by default. The simulation results show the correctness of the theoretical analysis and reveal that the distance-dependent Rician factor would transform the channel from LoS to NLoS when the transmission distance is increasing and vice versa; the outage probability would also be affected by different distance-dependent Rician factors; and although the DS cannot be removed when the direct link exists, it can be aligned to the same value. Thus, the Doppler spread can be eliminated. Besides, all results are obtained by using a personal computer with a 3.2 GHz AMD Ryzen 7 6800H CPU and 16 GB RAM. The software environment used is MATLAB R2020a.

**Table 2.** Simulation parameters [12,20].

| Parameters | Values |
|---|---|
| BS position ($D_{\mathrm{BS}}$) | $[0, 20, 50]^{\mathrm{T}}$ meter (m) |
| MR position ($D_{\mathrm{MR}}(t)$) | $[-250 + vt, 2, 20]^{\mathrm{T}}$ m |
| RIS height ($r$) | 15 m |
| Speed of the vehicle ($v$) | 180 km/h |
| Transmit power ($P_t$) | 20 dBm |
| Additive noise ($\sigma^2$) | $-80$ dBm |
| Carrier frequency ($f_c$) | 2.4 GHz |
| Element size ($dx$, $dy$) | $\lambda/2$ m, $\lambda/2$ m |
| Number of elements ($M$) | $64^2$ |
| Environmental parameters ($\varrho, \iota$) | 13, 0.03 |
| SNR threshold ($\gamma_{\mathrm{th}}$) | 10 dB |
| Realization number | $5 \times 10^3$ |

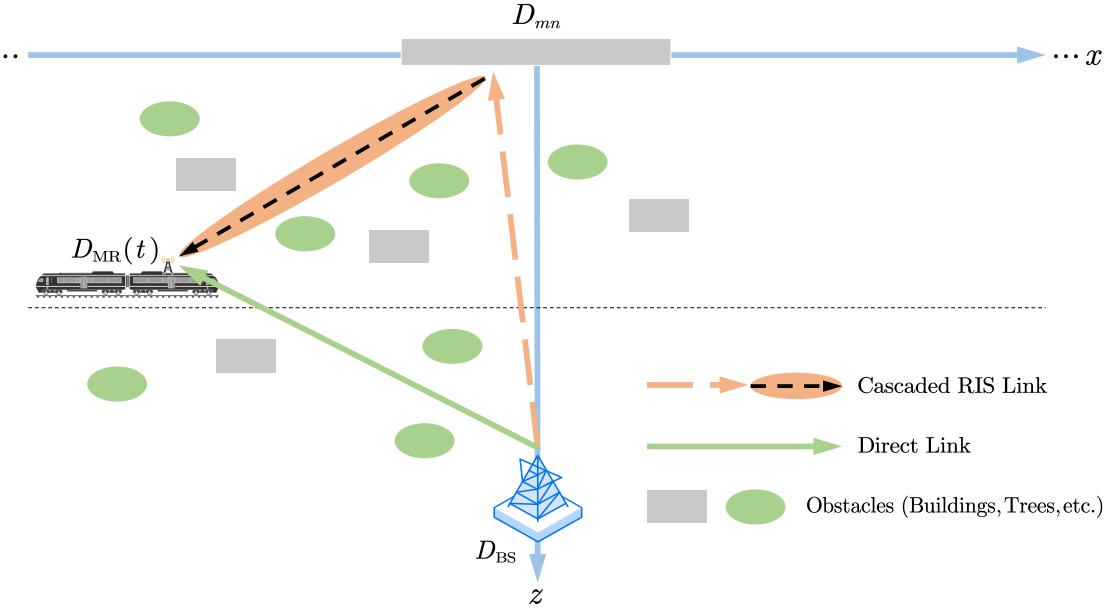

**Figure 2.** Top view of the proposed HSC system for numerical evaluations in this section.

### 5.1. Ergodic Spectral Efficiency

Figure 3 shows that the maximized ergodic SE of the proposed RIS-aided HSC system $SE^{upper}(t)$ in (29) increases from 5.41 bit/s/Hz to 11.79 bit/s/Hz, when the vehicle moving distances are 100 m and 250 m, respectively. This is because $d_0(t)$ and $d_{mn}^{MR}(t)$ are the smallest when the moving distance of the vehicle is 250 m. Then, the SE decreases to 5.41 bit/s/Hz again at 400 m moving distance, which is as expected since the channel gains $A_0(t)$ and $A_{mn}(t)$ and Rician factors are all distance-dependent. Therefore, the ergodic SE and the transmission distances (i.e., $d_0(t)$, $d_{BS}^{mn}$ and $d_{mn}^{MR}(t)$) are inversely proportional. Besides, for ease of comparison, we also provide the case without phase optimization, i.e., $\phi_{mn}(t) \in \mathcal{U}(0, 2\pi)$ where $\mathcal{U}$ is the uniform distribution. It is shown that before optimization, the ergodic SE of the proposed system is almost the same as the scenario without the RIS. This is because the power of the received signal increases only proportional to $M$, as (33) shows. Thus, obtaining the optimal phase shift $\phi_{mn}^{opt}(t)$ in (23) is crucial for the proposed system. Moreover, note that the analytical and simulated results of the ergodic SE match well, which validates Proposition 1. Therefore, we use the analytical upper bound SE expression in (29) in the rest of this section.

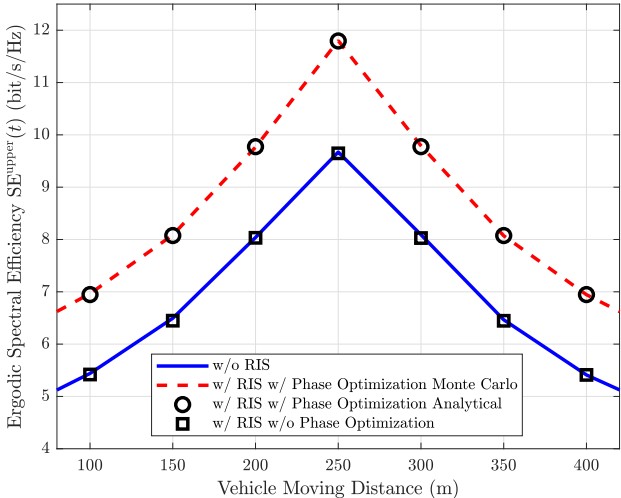

**Figure 3.** Ergodic SE for one vehicle pass with the distance-dependent Rician factors.

In order to validate Remarks [2], [6] and [7], we consider the MR position as $[-5000 + vt, 15, 20]^T$ m. It can be observed from Figure [4] that when the vehicle is far from the BS and the RIS, i.e., the vehicle moving distance is equal to or smaller (larger) than 1500 m (3500 m), the channel $h(t)$ is the same as the Rayleigh channel. However, when the $d_0(t)$ and $d_{mn}^{MR}(t)$ are small enough, i.e., the vehicle moving distance range is from 2350 m to 2650 m, $h(t)$ changes to the LoS channel. This result reveals that fixed Rician factors are not practical when describing the practical channel in the HSC system.

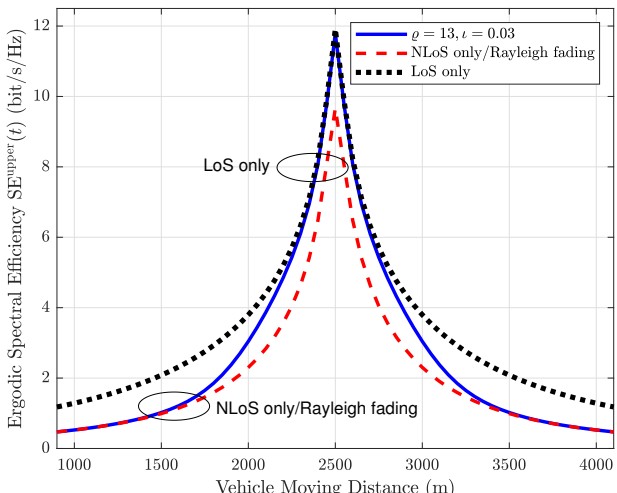

**Figure 4.** Ergodic SE for one vehicle pass with distance-dependent Rician factor ($\varrho = 13$, $\iota = 0.03$) [20], LoS Rician factor ($\varrho = 10^2$, $\iota = 0$), and NLoS Rician factor ($\varrho = -10^2$, $\iota = 0$).

Figure [5] shows the ergodic SE of the cases with different environmental factors $\varrho$ and $\iota$. It can be observed that the ergodic SE of the cases with $\varrho = 13$, $\iota = 0.03$ [20] and $\varrho = 0$, $\iota = 0$ are only the same when the vehicle moving distance is 1975 m. Other two of the same points appear in 2175 m and 2350 m, if $\varrho = 5$, $\iota = 0$ and $\varrho = 10$, $\iota = 0$ [19]. This result reveals that there is a clear difference between distance-dependent and fixed Rician factors. Therefore, the simulation results in Ref. [19] could be improved since it only considered the fixed Rician factors.

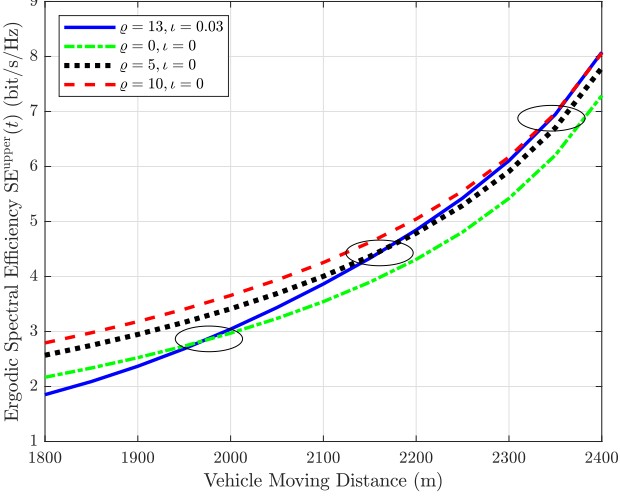

**Figure 5.** Ergodic SE for one vehicle pass with the distance-dependent Rician factor ($\varrho = 13$, $\iota = 0.03$) [20] and fixed Rician factors ($\varrho = 0$, $\iota = 0$; $\varrho = 5$, $\iota = 0$; $\varrho = 10$, $\iota = 0$ [19]). The moving distance of the vehicle is from 1800 m to 2400 m.

It can be observed from Figure 6 that from 2450 m to 2550 m, the case with dynamic Rician factors has the highest ergodic SE, which reveals that when the vehicle is near the BS and the RIS, the channel is almost the same as the LoS channel, rather than still with fixed Rician factors (e.g., $\varrho = 10$, $\iota = 0$ [19]). From Figures 5 and 6, it can be observed that the SE performance is overestimated (underestimated) when the transmission distance increases (decreases) in Ref. [19].

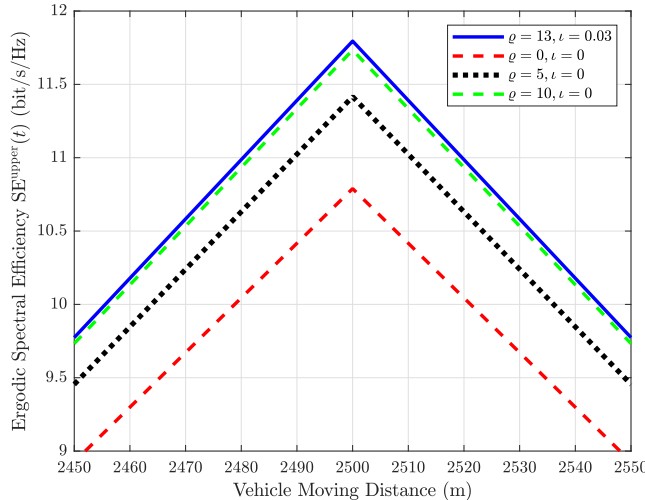

**Figure 6.** Ergodic SE for one vehicle pass with distance-dependent Rician factor ($\varrho = 13$, $\iota = 0.03$) [20], and Fixed Rician factors ($\varrho = 0$, $\iota = 0$; $\varrho = 5$, $\iota = 0$; $\varrho = 10$, $\iota = 0$ [19]). The moving distance of the vehicle is from 2450 m to 2550 m.

*5.2. Doppler Shift*

In Figure 7, we plot $f_0(t)$ and the $f_{mn}(t)$ with the maximal absolute value as the example. It can be seen see that after optimization (i.e., using $\phi_{mn}^{\text{opt}}(t)$ in (23)), the DS at time $t$, $f_{mn}(t)$, can be aligned to $f_0(t)$, which means that the Doppler spread is reduced to zero. Specifically, when the moving distance is 400 m, $\max\{|f_{mn}(t)|\}$ without optimization is 380 Hz, but $\max\{|f_{mn}(t)|\}$ with optimization is reduced to 295 Hz, the same as $f_0(t)$. Similarly, when the moving distance is 700 m, $\max\{|f_{mn}(t)|\}$ without optimization is 400 Hz, and $\max\{|f_{mn}(t)|\}$ with optimization is just 365 Hz. This result validates Remark 3. Besides, it can be observed that when the vehicle is far enough from the BS and the RIS, i.e., the moving distance is smaller (larger) than 100 m (900 m), the $f_0(t)$ and $\max\{|f_{mn}(t)|\}$ are nearly the same and stationary. This is because $d_0(t) \approx d_{\text{BS}}^{mn} + d_{mn}^{\text{MR}}(t)$ at this time. Besides, it is noteworthy that the positive and negative signs of the DS only reflect its deviation from the carrier frequency $f_c$. In other words, the positive and negative DS has the same influence on the system.

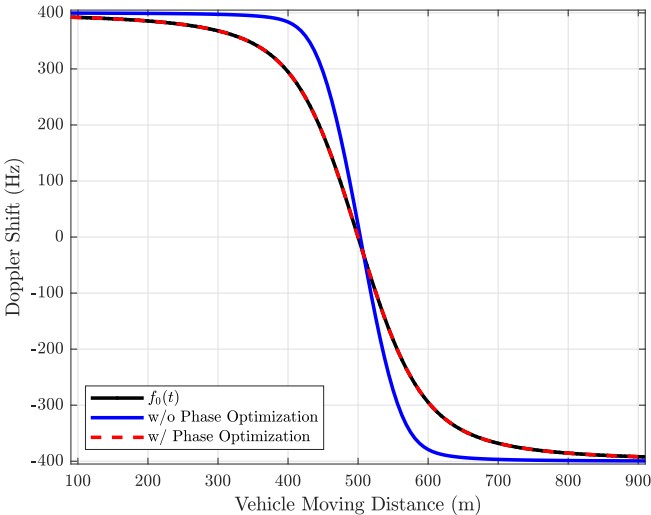

**Figure 7.** DS for one vehicle pass. $D_{\mathrm{BS}}$ is $[0, 20, 100]^{\mathrm{T}}$ (m), and $D_{\mathrm{MR}}(t)$ is $[-500 + vt, 15, 20]^{\mathrm{T}}$ m.

*5.3. Outage Probability*

From Figure 8, it can be seen that when the vehicle approaches the BS and the RIS, the outage probability reduces. Besides, increasing the element number of the RIS can decrease the outage probability. In particular, when $M = 64^2$, $128^2$, and $192^2$, the lowest outage probabilities are $6.05 \times 10^{-3}$, $5.70 \times 10^{-3}$ and $4.62 \times 10^{-3}$, respectively. This result validates Propositions 2 and 3. Besides, increasing the number of elements could decrease the outage probability, as expected, but more elements may bring a higher computational complexity if there is no prior channel state information. Therefore, a tradeoff between enhancing system performance and reducing computational complexity is necessary when designing a practical RIS-assisted HSC system.

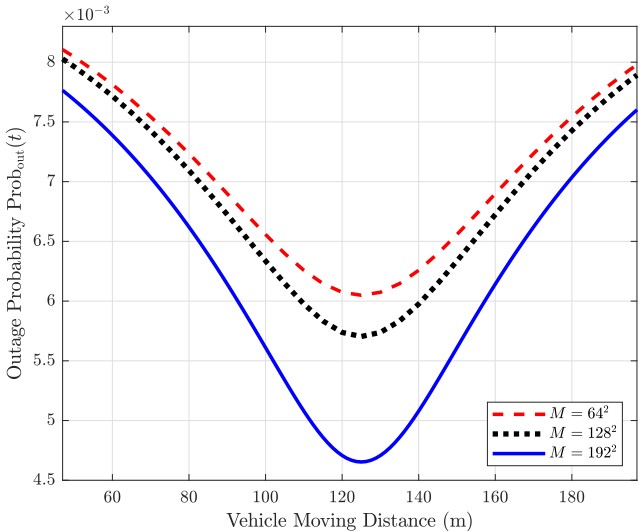

**Figure 8.** Outage probability for one vehicle pass. $D_{\mathrm{MR}}(t)$ is $[-125 + vt, 15, 20]^{\mathrm{T}}$ m.

### 5.4. Deployment Analysis

Figure 9 illustrates ergodic SE against different positions of the BS, i.e., $z_{BS} = 30$ m, 50 m, and 70 m. It can be seen that $z_{BS} = 30$ m can achieve the highest SE, as expected. This is because the path losses between the MR and the BS and between the RIS and the BS all decrease. Similarly, as Figure 10 shows, the outage probabilities of different $z_{BS}$s have the same pattern as Figure 9, due to the reduction of $d_0(t)$ and $d_{BS}^{mn}$. However, Figure 11 reveals that decreasing $z_{BS}$ would increase the optimized DS. Therefore, we should make a tradeoff between ergodic SE and DS, when deploying the BS. Specifically, deploying the RIS near the BS can enhance the SE and reduce the outage probability, at the expense of increasing the DS. Moreover, it should be emphasized that the negative sign of the DS in Figures 7 and 11 only denotes the level of the deviation, rather than representing that there exists a negative frequency in reality.

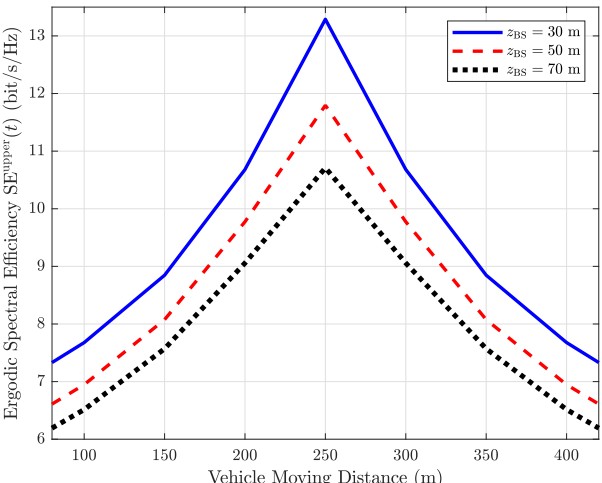

**Figure 9.** Ergodic SEs for different positions of the BS.

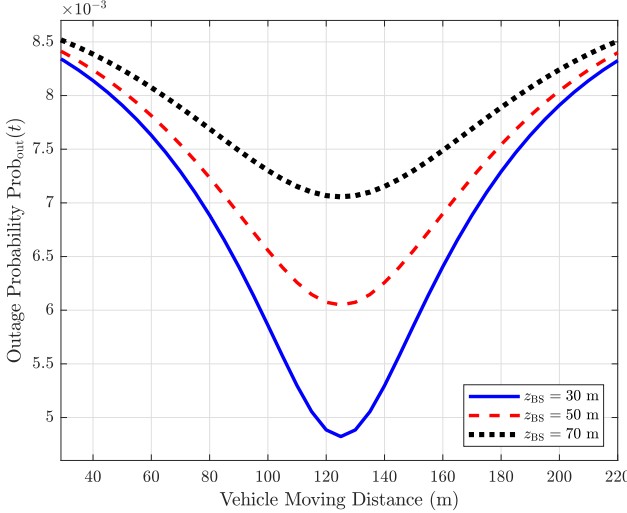

**Figure 10.** Outage probabilities for different positions of the BS.

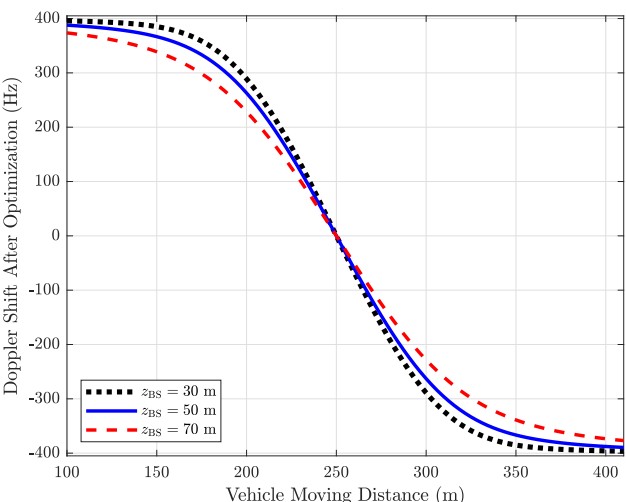

**Figure 11.** Optimized DS for different positions of the BS.

## 6. Conclusions and Future Works

In this work, we have proposed a time-varying channel model for the RIS-assisted HSC with the distance-dependent Rician factor. Compared to previous works, our model includes the Rayleigh fading component, the DS term, and the distance-related Rician factor, which is used to characterize the time-varying channel features. Using the proposed channel model, we have shown that the NLoS part dominates the channel when the moving user vehicle is far from the BS and the RIS, while the LoS part dominates when they are close. We have also shown that using RIS phase shift optimization, the DS in the cascaded links can be aligned to the shift in the direct link. Furthermore, if the direct link is obstructed, it is possible to completely eliminate the DS. We have derived the closed-form expressions for ergodic SE and outage probability of the proposed system. Besides, we have shown that the deployment strategy also affects the system performance.

For future works, one promising research direction is to optimize the location of the BS and the RIS to achieve the tradeoff between ergodic SE and DS. Other possible directions include measuring the parameters of the dynamic Rician factor from a practical environment, performance evaluation for a more practical scenario with non-isotropic elements of the RIS, and using channel aging techniques [23] to characterize the relationship between SE and DS.

**Author Contributions:** Conceptualization, K.W., C.-T.L. and B.K.N.; Methodology, K.W. and C.-T.L.; Software, K.W. and C.-T.L.; Validation, K.W. and C.-T.L.; Formal analysis, K.W. and C.-T.L.; Investigation, K.W.; Writing—original draft, K.W.; Writing—review & editing, K.W., C.-T.L. and B.K.N.; Visualization, K.W., C.-T.L.; Supervision, C.-T.L. and B.K.N.; Project administration, C.-T.L. and B.K.N. All authors have read and agreed to the published version of the manuscript.

**Funding:** This research received no external funding.

**Institutional Review Board Statement:** Not applicable.

**Informed Consent Statement:** Not applicable.

**Data Availability Statement:** Not applicable.

**Conflicts of Interest:** The authors declare no conflict of interest.

## Appendix A. Proof of Proposition 1

Consider $\mathbb{E}\{\log_2(1 + x)\} \leq \log_2\{1 + \mathbb{E}\{x\}\}$ [21], we then focus on the derivation of $\mathbb{E}\{|h(t)|^2\}$. We define $\Delta_1$, $\Delta_2$, $\Delta_3$ and $\Delta_4$ as follows

$$h(t) = \underbrace{A_0(t)K_1(t)h_d^{\text{LoS}}(t)}_{\Delta_1} + \underbrace{\sum_{m=1,n=1}^{\sqrt{M},\sqrt{M}} A_{mn}(t)\exp\left(-\jmath\phi_{mn}(t)\right)K_3(t)h_{mn}^{\text{LoS}}(t)G_1 g_{mn}^{\text{LoS}}}_{\Delta_2} +$$

$$\underbrace{A_0(t)K_2(t)h_d^{\text{NLoS}}(t)}_{\Delta_3} + \underbrace{\sum_{m=1,n=1}^{\sqrt{M},\sqrt{M}} A_{mn}(t)\exp\left(-\jmath\phi_{mn}(t)\right)\Big(K_3(t)h_{mn}^{\text{LoS}}(t)G_2 g_{mn}^{\text{NLoS}}(t) +}_{} \tag{A1}$$

$$\underbrace{K_4(t)h_{mn}^{\text{NLoS}}(t)G_1 g_{mn}^{\text{LoS}} + K_4(t)h_{mn}^{\text{NLoS}}(t)G_2 g_{mn}^{\text{NLoS}}(t)\Big)}_{\Delta_4}.$$

Note that $\Delta_1$ and $\Delta_2$ are deterministic, and $\mathbb{E}\{\Delta_3\} = \mathbb{E}\{\Delta_4\} = 0$. Therefore,

$$\begin{aligned}
\mathbb{E}\{|h(t)|^2\} &= \mathbb{E}\{h^*(t)h(t)\} \\
&= \mathbb{E}\{(\Delta_1 + \Delta_2 + \Delta_3 + \Delta_4)^* \cdot (\Delta_1 + \Delta_2 + \Delta_3 + \Delta_4)\} \\
&= \mathbb{E}\{\Delta_1^*\Delta_1 + \Delta_2^*\Delta_2 + \Delta_3^*\Delta_3 + \Delta_4^*\Delta_4 + \Delta_1^*\Delta_2 + \Delta_1\Delta_2^*\},
\end{aligned} \tag{A2}$$

where $(\cdot)^*$ denotes conjugate operation. Firstly, we focus on $\mathbb{E}\{\Delta_1^*\Delta_1\}$, which can be computed as

$$\mathbb{E}\{\Delta_1^*\Delta_1\} = \mathbb{E}\{\left(A_0(t)K_1(t)h_d^{\text{LoS}}(t)\right)^* A_0(t)K_1(t)h_d^{\text{LoS}}(t)\}. \tag{A3}$$

Recall that $h_d^{\text{LoS}}(t) = \exp\left(-\jmath 2\pi(f_c + f_0(t))\tau_0\right)$, we obtain

$$\mathbb{E}\{\Delta_1^*\Delta_1\} = K_1^2(t)A_0^2(t). \tag{A4}$$

We then concentrate on $\mathbb{E}\{\Delta_2^*\Delta_2\}$. Since the optimal phase shift set (23) is achieved, then the phase of the cascaded and direct links are the same, i.e.,

$$\underbrace{\exp\left(-\jmath\phi_{mn}(t)\right)h_{mn}^{\text{LoS}}(t)g_{mn}^{\text{LoS}}}_{\text{The phase of the cascaded link}} = \underbrace{\exp\left(-\jmath 2\pi(f_c + f_0(t))\tau_0\right)}_{\text{The phase of the direct link}}. \tag{A5}$$

Thus, we have

$$\begin{aligned}
\mathbb{E}\{\Delta_2^*\Delta_2\} &= \left|\exp\left(-\jmath 2\pi(f_c + f_0(t))\right)\right|^2 \left(\sum_{m=1,n=1}^{\sqrt{M},\sqrt{M}} A_{mn}(t)K_3(t)G_1\right)^2 \\
&= K_3^2(t)G_1^2 \left(\sum_{m=1,n=1}^{\sqrt{M},\sqrt{M}} A_{mn}(t)\right)^2.
\end{aligned} \tag{A6}$$

Next, we study $\mathbb{E}\{\Delta_3^*\Delta_3\}$. Recall that $h_d^{\text{NLoS}}(t) \sim \mathcal{CN}(0,1)$, thus $\mathbb{E}\{|h_d^{\text{NLoS}}(t)|^2\} = 1$, then we have

$$\mathbb{E}\{\Delta_3^*\Delta_3\} = A_0^2(t)K_2^2(t)\mathbb{E}\{|h_d^{\text{NLoS}}(t)|^2\} = K_2^2(t)A_0^2(t). \tag{A7}$$

Fourthly, for $\mathbb{E}\{\Delta_4^*\Delta_4\}$, we notice that $\mathbb{E}\{|h_{mn}^{\text{NLoS}}(t)|^2\} = \mathbb{E}\{|g_{mn}^{\text{NLoS}}(t)|^2\} = 1$, $\mathbb{E}\{h_{mn}^{\text{NLoS}}(t)\} = \mathbb{E}\{g_{mn}^{\text{NLoS}}(t)\} = 0$, hence $\mathbb{E}\{\Delta_4^*\Delta_4\}$ can be obtained as

$$\mathbb{E}\{\Delta_4^*\Delta_4\} = \left(K_3^2(t)K_4^2(t) + K_4^2(t)G_1^2 + K_4^2(t)G_2^2\right)\sum_{m=1,n=1}^{\sqrt{M},\sqrt{M}} A_{mn}^2(t). \tag{A8}$$

Finally, since $\Delta_1$ and $\Delta_2$ have the same phase part, then

$$\mathbb{E}\{\Delta_1^*\Delta_2\} = \mathbb{E}\{\Delta_1\Delta_2^*\} = A_0(t)K_1(t)K_3(t)G_1 \sum_{m=1,n=1}^{\sqrt{M},\sqrt{M}} A_{mn}(t). \tag{A9}$$

Therefore, considering all the results above, we have (30), which completes the proof.

## Appendix B. Proof of Proposition 2

Recall (A1), for a given time $t$, $\Delta_1$ and $\Delta_2$ are deterministic, which implies $\mathbb{E}\{\Delta_1\} = A_0(t)K_1(t)h_d^{\text{LoS}}(t)$, $\mathbb{E}\{\Delta_2\} = \sum_{m=1,n=1}^{\sqrt{M},\sqrt{M}} A_{mn}(t)\exp\big(-\jmath\phi_{mn}(t)\big)K_3(t)G_1h_{mn}^{\text{LoS}}(t)g_{mn}^{\text{LoS}}$, $\mathbb{V}\{\Delta_1\} = \mathbb{V}\{\Delta_2\} = 0$. Similarly, based on the definitions of (18)–(20), we have $\mathbb{E}\{\Delta_3\} = \mathbb{E}\{\Delta_4\} = 0$, $\mathbb{V}\{\Delta_3\} = A_0^2(t)K_2^2(t)$ and $\mathbb{V}\{\Delta_4\} = K_4^2(t)G_2^2 \sum_{m=1,n=1}^{\sqrt{M},\sqrt{M}} A_{mn}^2(t)$. The proof is then finished.

## Appendix C. Proof of Proposition 3

It has already proven in Proposition 2 that $h(t)$ follows a complex Gaussian distribution with expectation $\mu_h(t)$ and variance $\sigma_h^2(t)$. Thus, we have

$$\frac{|h(t)|^2}{\sigma_h^2(t)} \sim \chi^2(\nu, \xi(t)), \tag{A10}$$

where $\chi^2(\cdot)$ denotes non-central chi-squared distribution with degrees of freedom $\nu = 1$ and non-centrality parameter $\xi(t) = |\mu_h(t)|^2/\sigma_h^2(t)$. With the corresponding cumulative distribution function, outage probability $\text{Prob}_{\text{out}}(t)$ is given by

$$\text{Prob}_{\text{out}}(t) = 1 - Q_{\frac{1}{2}}\left(\sqrt{\xi(t)}, \sqrt{\gamma_0(t)}\right), \tag{A11}$$

where

$$\gamma_0(t) = \frac{\gamma_{\text{th}}}{\bar{\gamma}\sigma_h^2(t)}, \tag{A12}$$

and $\gamma_{\text{th}}$ is a given SNR threshold, $\bar{\gamma} = P_t/\sigma^2$. Note that $Q_u(a,b)$ is the Marcum Q-function [24], i.e.,

$$Q_u(a,b) = \frac{1}{a^{u-1}} \int_b^\infty x^u \exp\big(-\frac{x^2+a^2}{2}\big)I_{u-1}(ax)dx, \tag{A13}$$

where $I_{u-1}$ is the modified Bessel function of the first kind of order $u-1$.

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
