# Peer review of "RIS-Assisted High-Speed Communications with Time-Varying Distance-Dependent Rician Channels"

_applsci, doi:10.3390/app122211857_

Round 1
Reviewer 1 Report
In the research of reconfigurable intelligent surface (RIS), the author proposes a time-varying channel model with distance dependent Rician factor. The theoretical background and experimental process are analyzed and discussed. However, there are some details in the article that need to be modified.
1. Chapter 3, 4 and 5 suggest that the author briefly introduce the research content of this chapter before describing relevant content, so as to increase the readability of the article.
2. In the fifth chapter, the author conducted a lot of experimental analysis, and suggested that the author briefly introduce the development language or environment used in the experiment, so as to facilitate readers' reprint.
3. For section 5.3, it is recommended that the author move the text up to Figure 8 to increase the readability of the article.
4. In section 6, it is suggested that the author write down the current research contributions in sections, and then introduce the future work.
Author Response
Dear Editor and Reviewers,
Please see the .pdf file to find my response.
Thank you very much and best regards.
Ke"Ken"WANG

Reviewer 2 Report
Authors should discuss the effects of Doppler shift and spread on the system performance seperately. Is the amount of spread or shift is more influential on the system performance?
The authors need to clearly explain what they mean by tradeoff between ergodic SE and DS.
It would have been better to dicsuss the effect of gain of the reflected beam from the RIS on the SE and outage probability results
It would also be good if authors could provide SE versus Doppler shift and outage probability versus Doppler shift results
Author Response

(The authors gave the same response as above.)

Reviewer 3 Report
1) The order of equations referring in the manuscript are not in the order.
2) In Figure 7, authors have mentioned fmn(t) is -380Hz for w/o optimization, -295Hz for with optimization. What is the physical significance of –ve sign for frequencies?
3) In Figures 7 & 11, what is the significance of –ve values for Doppler shift?
4) In Section 6, the term “outrage probability” should be “outage probability”.
5) Literature review is very limited.
6) Authors are strongly recommended to do rigorous Literature review on the selected topic and compare the results of the proposed work with the existing Literature on Rician channel models.
7) The References are not cited in the order.
8) References [2], [4], [10], [11-13], [15-19] are mentioned with incomplete details.
9) English grammar needs major corrections.
Author Response

(The authors gave the same response as above.)
